# Stress Factors and Their Effects on Productivity in Sheep

**DOI:** 10.3390/ani13172769

**Published:** 2023-08-31

**Authors:** Hacer Tüfekci, Veerasamy Sejian

**Affiliations:** 1Department of Animal Science, Faculty of Agriculture, Yozgat Bozok University, Yozgat 66100, Turkey; 2Rajiv Gandhi Institute of Veterinary Education and Research, Kurumbapet, Pondicherry 605009, India; drsejian@gmail.com

**Keywords:** sheep, stress, welfare, reproduction, productivity

## Abstract

**Simple Summary:**

Stress factors (heat, cold, transport, treatment, nutritional, shearing, and weaning) are stimulants that initiate a stress response and are an inevitable result of today’s livestock practices. Stress factors, together with the changes in biological functions in animals, constitute the cost of stress. The animal’s response to a particular stress factor may vary depending on the animal’s breed, sex, age, and physiological status; the type of stress factor and its duration and density; and interactions between various factors, such as shelter, nutrition, climate, production practices, and the environment. This review aimed to reveal the importance of some stress factors in sheep and their effects on sheep productivity.

**Abstract:**

Products obtained from sheep have an economically important place in the world. Their adaptability to different climatic conditions, their ease of care and feeding, their high utilization of poor pasture areas with low yield and quality, the ease of flock management, their high twinning rate, and their short intergenerational period are some of the advantages of sheep production. Sheep production has the ability to adapt better to environmental stress factors, as can be understood from the presence of sheep in different geographical regions at a global level. However, the changes in environmental conditions and production cause some negative results in animals. All these negative results expose animals to various stress factors (heat, cold, transport, treatment, nutritional, shearing, weaning, etc.). All stress factors that directly and indirectly affect sheep production ultimately lead to compromised performance, decreased productivity, increased mortality, and adverse effects on the immune system. In order to cope with the current stress parameters in animals and to achieve optimum production, a holistic approach is needed according to the environmental conditions and available resources. It is important to consider the factors involved in these responses in order to manage these processes correctly and to develop adequate strategies and improve sheep welfare. This review aimed to reveal the importance of some stress factors in sheep and their effects on sheep productivity.

## 1. Introduction

Sheep production is a strategic branch of animal husbandry and has an important place in many countries, although sheep production varies according to market demand [1]. For many years, sheep have been raised by societies for their meat, milk, and wool due to their behavioral characteristics and high adaptability to the environment [2,3,4]. Their tail structures, one of the main features that distinguish sheep breeds, can be used as an energy store for the animals to cope with harsh environmental conditions [5,6,7,8]. This is one of the important advantages of sheep compared to other species.

Stress is a reflex reaction that occurs as a result of an animal’s inability to cope with the negative effects of various factors and its inability to adapt, and can have many negative results [9,10]. The stimuli that trigger stress are not necessarily painful, but psychological states such as fear or anxiety activate physiological responses that could impose a deleterious impact on the animal. When the animal perceives a threat, it develops a behavioral, autonomic, endocrine, or immune response to maintain homeostasis [11,12]. If the animal cannot withstand the stress, this may cause significant disturbances in the organ functions and systems of the body, abnormal biological functions, and pathologies. In addition to physiological changes under the influence of the stressor, this can lead to significant dysfunction and behavioral changes in the animal, and even death [11,13,14]. Stress responses are related not only to the nature, intensity, and duration of the triggering stimulus, but also to the individual response tendencies or temperament of the animals [11].

Factors that cause stress in farms and pastures mostly include unfavorable housing conditions, inappropriate procedures, veterinary procedures (treatments, vaccinations, blood tests, and surgical interventions), animal production practices (numbering, weaning, grouping, shearing, tail docking, and hoof care), adverse weather conditions (extreme heat and cold), and malnutrition [14,15]. In addition, the conditions that are provided during the pre-slaughter period for the animals to be slaughtered, transport vehicles, loading and unloading, transport, pre-slaughter feeding and watering, the treatment of the animal in the slaughterhouse, and stunning practices are also considered to be stress factors [16]. 

During the transportation of animals, poor road conditions, the driving performance of the drivers, transport vibrations, road length, climate, and the lack of water and feed can cause stress in animals [17,18,19]. When these stress factors are repeated, the intensity of the action increases; when multiple stresses act simultaneously, their negative effect on the animal will increase [14,20]. In addition, studies have also reported that stress in farm animals may have detrimental effects on the quality of food products [21,22] and may have potential effects on food safety [23].

All farm animals will experience some level of stress (psychological or physiological) at different periods throughout their lives [23,24]. If the exposure to stress factors in farm animals can be managed correctly, this will have a positive effect on the productivity of the animals and will also increase their welfare. Some stress factors in sheep production are shown in Figure 1, and some negative effects of the stress factors on sheep productivity are shown in Figure 2.

## 2. Cold Stress

For each species, the temperature values of the thermal comfort zone for optimum production are different, as well as the temperature sensitivity of the species varying according to their age and physiological periods. Most often in farm animals, the focus is on the negative effects of heat stress on animals rather than cold stress. However, in winter, one of the biggest stress factors for animals is exposure to cold. Cold stress increases energy expenditure in ruminants [25], resulting in increased utilization of lipids as an energy substrate [26]. In the case of cold stress, animals can compensate for their body temperature by increasing feed intake to meet their increased energy requirements for life support [27] and it is generally stated that cold weather has a minimal effect on the reproductive ability of animals [28]. However, one of the biggest problems in livestock is seasonality and nutritional status [29]. Feeding animals on inadequate pastures is an important factor that can negatively affect animal performance, and feed intake is an important factor that affects biological indicators in the body [30,31]. Ewes are seasonal polyestrous animals and births usually occur in winter. The effectiveness of feed utilization during this period can be poor. The reasons for this include low temperatures, which will lead to low food intake, low digestibility, and high requirements for maintaining body temperature [32,33,34]. Therefore, exposure to cold during winter in ruminants is defined as a source of stress that impairs their growth, production, and reproduction [35]. Furthermore, sheared ewes and lambs are particularly susceptible to cold exposure [36,37]. Sejian et al. (2017) [38] reported that cold stress negatively affected growth, physiological adaptation, blood metabolites, stress, and reproductive hormone levels in Malpura ewes. Simeonov et al. (2022) [34] reported that lambs reared at an average temperature of 12.6 °C gained significantly more live weight than lambs reared at lower temperatures in a study conducted to determine the effect of different environmental temperatures (medium: 12.6 °C; low: 5.1 °C; very low: −3 °C) on growth and feed intake in early weaned lambs. These reports highlight the negative impact of cold stress on sheep and thus appropriate mitigation measures need to be taken to reduce the losses and improve the animal welfare.

It has been reported that cold, wet, and windy weather may cause a significant increase in lamb mortality and hypothermia may be responsible for almost half of perinatal lamb losses [39,40,41]. Therefore, cold stress is a critical factor in the survival of newborn lambs [42]. Lambs lose heat quickly when wet, especially in windy and cold weather, and providing shelter and protection of newborn lambs from wet and wind chill can contribute significantly to reducing thermal stress [42,43]. Genetic variation in cold tolerance of newborn lambs is known to exist and recent research has shown that genetic variation exists in the β3-adrenergic receptor in Merino lambs; these variations are associated with cold tolerance and mortality from cold stress [42,44].

Environmental temperature is the main exogenous regulator of thyroid gland activity and there is an inverse relationship between ambient temperature and blood thyroid hormone (TH) concentrations in sheep and goats [45,46,47,48,49]. It has been reported that seasonal variations in blood TH levels generally show minimum values in summer and maximum values in winter months [50,51,52]. Souza et al. (2002) [53], according to results obtained from sampling every 2 months for 1 year, reported that rams showed the highest TH concentration in the afternoon and the lowest TH concentration in the early morning. Generally, lamb breeds reared in extensive conditions (hill regions) have better thermoregulation than those reared intensively in the lowlands, which is related to birth coat characteristics accompanied by higher concentrations of thyroid hormones (important for endogenous heat production and coat growth) in hill lambs than in lowland lambs [54]. Doubek et al. (2003) [13] reported that 2- to 3-day-old Merino lambs exposed to cold stress showed a stronger increase in thyroid hormone levels than Romney Marsh lambs. Dwyer and Morgan (2006) [55] reported that Blackface lambs had higher T3 and T4 levels than Suffolk lambs at birth, which were associated with the higher body temperature and better thermoregulation ability of Blackface lambs.

Studies in different species have reported that HSP70, which varies seasonally and temporally, can be used as a biomarker for adaptation to hot and cold stress conditions [56]. Banerjee et al. (2014) [57] investigated the seasonal variation in the expression pattern of genes under the HSP70 family in heat- and cold-adapted goats (*Capra hircus*) and reported that HSPA1A and HSPA8 expression was higher in winter months in both heat- and cold-adapted goats, but other HSPs were down-regulated. Furthermore, the results obtained indicate that the expression pattern of HSP70 genes is species-specific and breed-specific due to differences in thermal tolerance and adaptation to different climatic conditions. Mohanarao et al. (2014) [58] reported that HSP27 is a useful indicator for evaluating the stress response in vitro and provides the basis for further in vivo studies to validate this gene as a thermal stress indicator in animals. In addition, it was reported that this study constitutes an important step in understanding the cellular stress response in terms of the expression of HSP genes against heat and cold in goats.

## 3. Heat Stress

Economic losses associated with temperature stress in the livestock industry include slow growth rates, reduced fertility, increased veterinary costs, animal diseases, unfavorable milk quantity and quality, inconsistent carcass quality and composition, reduced market weights, and increased animal welfare issues [59,60]. Livestock are susceptible to heat stress due to their metabolic rate and growth, high production levels, and species-specific characteristics such as rumen fermentation, sweating, and skin insulation. Heat stress is the most important stress in the life of livestock with detrimental consequences for animal health, productivity, and product quality, thus directly affecting animal production. Moreover, due to its indirect effects, heat stress in summer severely hinders pasture and water availability, which ultimately results in severe nutritional and water stress in animals [60,61]. Although small ruminant farming is mostly pasture-based, changes in feed and water resources can cause stress. Under extreme environmental conditions, both the quantity and quality of available pastures are affected [62]. With the changing climate, animals may have to walk long distances to find pasture, and this movement causes great stress in animals. Thus, animals are exposed not only to temperature stress, but also to feeding, water, and walking stress. In this case, the animal’s body reserves will not be sufficient to effectively counteract multiple environmental stressors, as the animal will be exposed to multiple stressors at the same time. Consequently, this is attributed to the animal’s inability to cope with the combined effects of different stressors, the animal cannot adapt and struggles to maintain normal homeothermia [61,63,64]. It has been reported that water requirements of heat-stressed animals can double during heat stress compared to heat-neutral conditions [65]. The increase in water intake is crucial for thermoregulation to counteract water evaporation through panting and sweating [66]. In addition, water is an essential nutrient required for many physiological functions necessary for maximum performance in livestock [67]. Water is necessary for body temperature regulation, growth, reproduction, lactation mechanisms, the digestion model, nutrient exchange, excretion of waste products, and heat balance. Studies have reported that exposure of ewes to hot environmental conditions causes a significant increase in water intake as well as water cycling [68,69]. 

High temperature causes physiological, biochemical, and behavioral changes that negatively affect the welfare of sheep, which in turn negatively affects the immune, nervous, and endocrine systems of the animals, increases the health problems of the animal, and negatively affects parameters such as animal life span, quality, and yield level. Furthermore, stressful situations caused by adverse environmental conditions can also significantly reduce fiber diameter, wool quality, and fiber strength [70,71]. Since meteorological characteristics such as relative humidity, solar radiation, and wind speed together with high ambient temperature increase the degree of sensation of environmental temperature, these data should be evaluated as a whole. This situation may cause critical effects of temperature stress on the yields obtained in animal production in farms [72,73,74]. 

### 3.1. Impact of Heat Stress on Physiological Profile

The main consequence of heat stress in livestock is a reduction in feed intake, which is a natural, protective, and adaptive mechanism used by animals under high ambient temperatures to counteract the increase in metabolic heat production [75,76]. Verma et al. (2000) [77] reported in their study that changes in the concentration of thyroid hormones in the blood reflect the metabolic and nutritional status of the animal, and that decreased feed intake was due to the direct effect of increased temperature on the satiety center of the hypothalamus. In addition, it has been stated that heat stress increases the secretion of leptin and adiponectin in animals, causing a decrease in feed consumption. Slimen et al. (2019) [78] reported that environmental temperature stress affects not only physiological responses but also the energy and oxidative metabolism of sheep exposed to environmental temperature stress, that decreased feed intake is partly responsible for decreased energy metabolism, and oxidative stress is also responsible for metabolic disorders that further alter ovine production performance. Rana et al. (2014a) [79] conducted a study to investigate the effect of heat stress on blood parameters in sheep and divided the animals into three groups as zero hours (T0), four hours (T4), and eight hours (T8) of exposure to direct sunlight. As a result of the study, it was reported that red blood cell (RBC), hemoglobin (Hb%), and packed cell volume (PCV%) increased significantly with increasing temperature stress and that temperature stress caused significant changes in some blood parameters in domestic sheep.

### 3.2. Impact of Heat Stress on Sheep Milk Production

It has been reported that milk production characteristics in ovine animals are negatively correlated with temperature or relative humidity, different sheep breeds have variable tolerance to temperature and humidity, and sunlight has less effect on milk yield and more effect on casein yield, fat, and clot hardness [11,80,81]. It is reported that increasing the environmental temperature jeopardizes udder health and milk quality in ewes and may lead to mastitis or lower milk quality. Furthermore, exposure to intense solar radiation adversely affects the hygienic quality of milk. The increased bacterial load in milk is represented by higher amounts of pathogenic microorganisms and milk neutrophils and increases the somatic cell count [82]. High environmental temperatures can increase energy requirements by up to 25%, jeopardizing productivity in lactating ewes. Both heat stress and the progression of lactation result in decreased mobilization of body reserves for milk synthesis and reduced milk yield and quality. In studies, it has been reported that heat stress decreases both the fat and protein content of milk and in parallel with the increase in temperature and humidity index, milk yield decreases up to 75% [83].

### 3.3. Impact of Heat Stress on Sheep Growth, Meat and Carcass Traits

Exposure of sheep to high temperatures causes a decrease in body weight, daily body weight gain, growth rate, and total body weight [84]. As a result of adaptive responses to chronic heat stress, animals generally reduce feed intake to reduce metabolic heat production, which has effects on carcass fat accumulation, carcass yield, and intramuscular fat content [66]. In addition, ambient temperature is reported to affect feed digestibility by changing the volume of the gastrointestinal tract and the rate of passage through the digestive tract [67,85]. Recent studies have shown that heat stress not only causes physiological and metabolic disturbances in animals, but also may affect carcass and meat quality characteristics by altering the rate and degree of postmortem muscle glycolysis and the resulting pH [86].

Heat stress is known to be associated with dark-colored firm and dry meat (DFD) in ruminants [87]. Animals exposed to chronic heat stress have reduced muscle glycogen reserves, leading to lower lactic acid production, resulting in dark, tough, and dry meat, characterized by a high final pH and greater WHC, which is commonly observed in ruminant animals [88,89]. Rana et al. (2014b) [90] reported that heat stress was effective on carcass characteristics and meat quality traits of domestic sheep and temperature stress decreased the weight of edible and non-edible components. Kadim et al. (2008) [73] showed that high ambient temperatures caused significant negative effects on meat quality in sheep and goats. Sheep and goats slaughtered at an ambient temperature of 35 °C had higher pH values and lower color (lightness, redness, and yellowness) than those slaughtered at 21 °C, indicating that seasonal temperatures are the main cause of differences in meat quality. In addition, it has been reported that meat shelf-life and safety decreases in various species due to increased lipid and protein oxidation and a favorable environment for bacterial growth during hot seasons [91,92].

### 3.4. Heat Stress and Sheep Reproduction

Heat stress potentially reduces mating behavior and fertility of ewes and rams, increases embryonic mortality, impairs embryo development, reduces lamb birth weight, and increases perinatal lamb mortality [93,94]. Studies have shown that heat stress reduces behavioral estrus duration, estrus incidence, and cycle length [95,96,97]. Heat stress impairs sperm production and sperm quality in rams and negatively affects semen quality by increasing the rate of morphologically abnormal sperm [98,99,100]. In addition, heat stress also negatively affects prenatal and postnatal offspring development in ewes. Continuous exposure of ewes to high ambient temperatures during pregnancy impairs placental development and fetal growth, resulting in the birth of lambs with high probability of death or low birth weight [101,102,103]. Low birth weight is one of the main risk factors contributing to neonatal lamb mortality [104,105]. In summary, the reproductive performance of ewes is adversely affected by increasing temperatures and heat stress makes ewes more vulnerable to reduced reproduction, production, milk quantity and quality, meat quantity and quality, as well as reduced immunity and even offspring mortality.

## 4. Nutritional Stress

Although sheep farming is practiced in almost all regions of the world, it is widespread in arid and semi-arid regions. Nutritional stress is a major constraint to ruminant livestock production in semi-arid regions. With the cessation of rainfall, the quantity and quality of grazing land declines rapidly, leaving cereal crop residues as the main feed source. These residues are low in nitrogen (N) and high in crude fiber; these characteristics limit intake and digestibility, resulting in malnutrition [106]. Sheep grazing in these areas face extreme fluctuations in the quantity and quality of feed offered throughout the year. It has been reported that the energy demands of animals vary according to different seasons and physiological periods and if ewes are exposed to prolonged nutritional stress, it may cause a wide range of negative effects on their productivity, affecting both immediate production and lifetime performance [107,108,109].

### 4.1. Impact of Nutritional Stress on Physiological Status of the Animal

Pregnancy is a physiological process characterized by a significant increase in energetic and oxygen demands to ensure adequate fetal growth and development. Therefore, both mother and fetus may experience oxidative stress during pregnancy. The nutritional levels of pregnant animals also affect their energy reserves [27,110]. Nawito et al. (2016) [111] reported that feeding under arid conditions caused oxidative stress in pregnant sheep and goats. Maternal nutrition is a factor that can adversely affect offspring through altered macro- or micronutrient intake during pregnancy. Both maternal nutrient restriction and overfeeding result in offspring that exhibit poor postnatal growth, reduced muscle mass, increased adiposity, impaired metabolism, and altered immunity. Furthermore, offspring born to restricted-fed and overfed mothers are at risk of developing chronic metabolic diseases in adulthood [112,113,114,115]. Sejian et al. (2014) [116] investigated the effects of dietary restrictions on growth, physiological adaptation, and reproduction in sheep. As a result, 60% dietary restriction significantly decreased body weight, body condition score, plasma insulin-like growth factor, pulse rate, respiratory rate, plasma glucose, and significantly increased total cholesterol, plasma tri-iodo-thyroxine and thyroxine, plasma cortisol, estrus % and estrus duration, plasma concentration of growth hormone, plasma concentration of progesterone, and estrous cycle length. Parraguez et al. (2022) [117] examined the effects of antioxidants and nutritional supplementation on early oxidative stress and fetal growth restriction in twin pregnancies of malnourished ewes. As a result, it was reported that, in addition to nutritional supplementation, the administration of herbal antioxidants may constitute a good nutritional strategy for ewes reared under harsh environmental conditions. Trotta et al. (2020) [118] conducted a study to investigate the effects of nutrient restriction from mid to late gestation on maternal and fetal digestive enzyme activities in ewes and reported that some maternal and fetal digestive enzyme activities may change in response to maternal nutrient restriction. Maurya et al. (2019) [119] in their study to evaluate the effects of combined stresses (temperature and nutrition) on physiological adaptation, blood biochemical, and endocrine responses in Malpura rams; it was stated that when two stressors occur simultaneously, the total effect on the adaptive ability of rams to adapt to stressful conditions can be severe.

### 4.2. Impact of Nutritional Stress on Sheep Milk Production and Composition

Most of the sheep milk produced in the world is processed into cheese, yogurt, and other dairy products. Sheep milk composition is strongly influenced by nutrition, especially in high yielding animals. In particular, milk fat composition, milk protein content, milk flavor, and milk yield are easily influenced by the diet of the animals. In addition, nutritional stress and certain vitamins affect the somatic cell content of milk, which in turn affects cheese yield and quality. As dairy sheep are usually grazed, they are often subjected to nutritional stress due to variability in pasture characteristics and the influence of meteorological events. In general, the botanical composition of pastures and nutrition can modify the flavor, quality, and safety of sheep milk, and it is also possible to improve these characteristics by nutritional characteristics of the diet [120]. Al-Saiady (2006) [121] reported that the milk performance of ewes of two sheep breeds fed diets with normal and limited dry-matter intake and different concentrate/roughage ratios were significantly affected. In addition, the decrease in milk yield under heat stress is greater for older and higher lactating animals, especially during the peak of lactation. There is also a decrease in milk yield in animals experiencing cold stress, but this decrease is less than in animals experiencing heat stress [122,123].

### 4.3. Nutritional Stress and Sheep Reproduction

Reproductive performance in ewes is strongly influenced by feed availability during the mating season. Idris et al. (2010) [124] tested different feed sources to examine their effects on ewe productivity and reproductive performance during two physiological stages (mating period and pre-lambing period) in ewes. As a result, they reported that nutritional supplementation improved body condition score, lambing rate, fertility rate, fecundity, pregnancy, weaning rate, and reduced abortion rate in ewes. Borowczyk et al. (2006) [125] reported that nutrition affects various reproductive functions including hormone production, oocyte competence, fertilization and early embryonic development. It was reported that malnutrition of ewes resulted in a low body weight and body condition score, which had a negative effect on oocyte quality, resulting in lower cleavage and blastocyst formation rates. Musa et al. (2018) [126] reported that vitamin E administration alone and in combination with selenium improved reproductive performance in ewes.

### 4.4. Impact of Nutritional Stress on Wool and Fiber Quality

Hunter et al. (1990) [127] investigated the effects of feeding and lambing stress on wool fiber and wool fiber properties in Merino, Dohne Merino, and Mutton Merino sheep. As a result, it was reported that limited feeding and lambing stress reduced fiber diameter by 30% on average. Fiber strength was not affected by feeding or lambing stress, but the observed differences in fiber strength were largely due to differences in fiber diameter. Olivier and Olivier (2005) [128] investigated the effect of short-term feeding stress after weaning on wool production characteristics of Merino ewes. As a result, they reported that nutritional stress inactivated the wool follicles of ewes in different groups and some follicles shed their fibers, but the inactivation of follicles was not permanent and did not have a permanent detrimental effect on the wool production potential of ewes.

In conclusion, nutrition is a fundamental factor affecting reproduction, fertility, survival, milk yield and quality, live weight, meat yield and quality, fleece yield and quality, and health in ewes, and it is also an application that can be used to reduce the effects of some stress factors. Therefore, short and long term exposure of animals to nutritional stress can cause many negative effects on metabolism, physiology and yields.

## 5. Transportation and Treatment Stress

Animal transport is a very important link in the supply chain and is also one of the most problematic areas of animal welfare. The standards that must be complied with for live animal exports, both for road and maritime transport, are of great importance in the livestock industry. No matter how well-organized animal transport is, a sudden change in living conditions on farms can lead to changes in animal welfare and animals trying to adapt to the current situation. This is expressed as a typical stress reaction that animals are exposed to during transport. In addition, poor road conditions, driving performance of drivers, transport conditions, road length, climate, lack of water and feed can cause stress in animals during transport [19,129,130]. Some studies have shown that the main factors affecting the welfare of sheep during road transport are driving behavior, noise, acceleration, road type [131,132], vehicle design [133,134], transport intensity [135], duration of the journey [136,137] and transport of different breeds together, new environment, hunger, thirst and fatigue [133], excessive ventilation, cold or heat stress [138,139]. It is recommended that transport vehicles should never be completely closed as lack of ventilation can cause extreme stress and even suffocation [140].

Fear, hunger, thirst, physical injury and fatigue due to pre-slaughter treatment, capture, loading, transport, unloading, and stunning techniques have negative effects on animal health, welfare, and meat quality [141,142,143]. The energy required for muscle activity in live animals is provided by muscle glycogen. In rested healthy animals, the glycogen content of muscles is high. After slaughter, blood circulation in the body is interrupted, so is the delivery of oxygen and nutrients to the muscles and muscle fibers. Metabolic processes take place in the absence of oxygen and muscle glycogen is converted into lactic acid. Since lactic acid is not transported with the circulation, it remains in the muscles and this causes the pH to drop from the normal values of 7.00–7.40 to 5.40–5.70 in live animals. One of the main criteria for meat quality is the final pH of the muscles, determined by glycogen depletion and post-mortem lactic acid accumulation. Lactic acid is essential to produce tender meat with a good taste and good color. If the animal experiences stress before and during slaughter, glycogen is depleted and the lactic acid formed in post-mortem meat is reduced. This has a serious negative effect on meat quality. Therefore, animals should not be exposed to stress and injuries during pre-slaughter procedures to prevent unnecessary depletion of muscle glycogen stores. Resting the animals in the 24-h pre-slaughter period is an important issue that should be emphasized in order to maximize muscle glycogen in the slaughtered carcasses and to create ideal conditions [144,145,146,147].

Kadim et al. (2006) [148] reported that the transport of animals had a significant effect on meat quality and caused a significant increase in meat pH and shear force values in animals transported before slaughter. In another study, lambs carried for 5 h had lower carcass yield and water-holding capacity of meat than lambs carried for 30 min [149]. Andronie et al. (2008) [129] reported that plasmatic glucose levels decreased, urea levels increased, body weight decreased due to travel time and lack of feed and water during this period, thus travel time is one of the main transport-related stress factors leading to a decrease in the quality level of sheep welfare. Lendrawati et al. (2020) [150] investigated 4, 8 and 12 h of road transport duration on body weight loss, hematological and biochemical responses of sheep. The results showed that increased transport time significantly increased body weight loss and decreased hematocrit, cortisol, and glucose values in blood. Tozlu Çelik et al. (2021) [19] aimed to determine the effects of transport and altitude in Karayaka sheep. The results obtained showed that transport and altitude significantly affected triiodothyronine, tyrosine, and oxidative stress parameters of malondialdehyde (MDA), so these factors were reported to cause stress in sheep. Ekiz et al. (2012a) [151] reported that in their study with different sheep breeds, significant breed differences were observed in terms of behavioral responses to some physiological, biochemical, and stress factors that may arise from transport and related transport procedures, and therefore, it was reported that breed differences in stress responses should be taken into account in the planning of transport and production procedures while considering an animal welfare approach. In conclusion, all animals are under a certain amount of stress during transport and all treatments prior to slaughter, which can be detrimental to meat yield and meat quality.

## 6. Shearing Stress

Shearing is an important adaptation strategy for many sheep breeds to seasonal changes, especially in summer, and it can also change the resistance of animals to high temperatures, but in recent years, shearing has not been carried out regularly on farms in some regions for various reasons. Gowane et al. (2017) [152] reported that the amount of wool produced from sheep varies according to climate region, breed, age, sex, nutrition, shearing range, and the commercial value of wool also depends on wool yield and quality. Therefore, shearing is a necessary practice in sheep production to ensure wool removal, usually on an annual basis. Shearing before the onset of hot weather can reduce the effects of heat stress in sheep. However, factors such as grabbing, turning, dragging, foot binding, separation from the flock, waiting time, and close human contact during shearing and exposure of animals to noise and possible skin injuries during shearing all constitute an important short-term stressor that can jeopardize animal welfare [153,154,155]. 

In some studies, it has been reported that shearing in different seasons significantly affects thermoregulation and some physiological and blood parameters [156,157], and also shearing affects oxidative parameters and causes a change in the homeostatic balance of sheep leading to oxidative stress [158]. Hargreaves and Hutson (1990) [159] reported that shearing is a routine husbandry practice and produces an acute short-term stressor as indicated by a significant increase in plasma cortisol concentrations. Panaretto (1968) [160] reported that shearing of sheep causes cold stress in animals for a period of time after shearing, depending on the climate and fleece length at the time, and post-shearing cold stress significantly increases heat loss and feed requirements, resulting in losses in production efficiency.

Piccione et al. (2002) [161] investigated the effects of shearing on body temperature in different sheep breeds in spring. As a result, shearing caused an increase of over 1 °C in the internal body temperature of sheep for all breeds, which was expressed as an overreaction to the mild cold stress caused by fleece loss. Elvidge and Coop (1974) [162] reported that there were significant differences in the results of their study in which the increased heat production and feed requirements resulting from shearing were estimated from the body weight changes in sheared and non-sheared ewe diets. Carcangiu et al. (2008) [163] investigated how stress caused by shearing stages can alter growth hormone (GH) cortisol and some hemato–chemical parameters in the Sarda sheep breed. The results obtained show that GH secretion in animals is affected by the whole stress procedure (separation, tying, and shearing). Even if shearing is necessary for animals, it was reported to cause a significant change in blood parameters involved in the stress response. Fidan et al. (2009) [164] determined the stress response of Sakız sheep during shearing and evaluated its effects on oxidant–antioxidant status. Blood malondialdehyde (MDA) and cortisol concentrations increased after shearing compared to baseline, while glutathione (GSH) concentration decreased significantly. These results suggest that MDA and GSH are potent markers for assessing oxidant/antioxidant status and indicate that shearing is a stressful situation leading to oxidative stress, which can be increased by strong glucocorticoid secretion. Al-Ramamneh et al. (2011) [165], investigated the effect of shearing on water turnover, body surface temperatures, and other physiological variables in sheep reared under summer temperate conditions. As a result, it was reported that shearing under temperate conditions significantly decreases body temperature, water intake, and respiratory rate and thus causes stress in sheep, which should be taken into account when determining shearing times under current management practices. Casella et al. (2016) [166] reported that thermal stress caused by shearing is one of the most stressful events in the life of sheep, considering that shearing can affect the thermoregulatory mechanisms and welfare of sheep. Arfuso et al. (2022) [153] investigated the usefulness of eye temperature assessment by infrared thermography (IRT) to evaluate the acute stress response in sheep during shearing and reported that IRT measurement allows a more accurate interpretation of the behavioral response in sheep. 

In conclusion, shearing is a necessary practice in sheep production, but it is a fact that the treatment that sheep are exposed to before, after, and during shearing negatively affects animal welfare and causes acute stress.

## 7. Weaning Stress

Weaning leads to a change in environmental conditions including the feeding regime for the offspring. Commercial farms usually separate ewes and lambs before natural weaning. Early weaning results in the breaking of the ewe–lamb bond, with changes occurring simultaneously in both the physical and social environment, coupled with the cessation of suckling and the complete replacement of milk with solid foods. This process varies depending on the objectives of each sheep farm and the characteristics of the lambs, including their age and weight, and their ability to eat solid food. Early weaning triggers behavioral, physiological, and immunological changes with negative consequences on the health of both ewes and lambs, as well as raising animal welfare concerns over lamb growth [167,168,169]. 

Freitas-de-Melo et al. (2013) [167] reported that sudden weaning, a common management practice in sheep production systems, causes behavioral and physiological responses indicative of stress in ewes and lambs. Schichowski et al. (2008) [170] investigated the responses to weaning in 50- and 100-day-old lambs and reported that weaning caused a significant increase in cortisol levels in both lamb groups.

One of the most important consequences of weaning stress in lambs is decreased feed intake. The decrease in feed intake leads to a decrease in growth rate and thus increased susceptibility to diseases [168]. Ekiz et al. (2012b) [171] investigated the effects of weaning duration and rearing type on average daily body weight gain (ADG) of lambs at different growth stages. The results obtained showed that weaning caused a decrease in ADG of lambs. Pascual-Alonso et al. (2015) [172] analyzed the effect of post-weaning treatment strategies on welfare and production traits of lambs. They reported that the use of social enrichment at weaning, i.e., gentle handling and daily human contact, resulted in lambs being less reactive and better able to regulate their physiological stress in the post-weaning period, thus improving lamb health and weight. Mohapatra et al. (2021) [173], in their study to predict early weaning-related stress in lambs, showed that comparable changes in body weight and blood biochemical levels indicated that weaning at 60 days enabled lambs to cope with behavioral and physiological weaning stress.

Napolitano et al. (2008) [174] reported that early weaning negatively affects the welfare of lambs, with clear and obvious detrimental effects on various functions of animals. In addition, they stated that behavioral measures are more sensitive for the detection of stressful conditions compared to endocrine or immune indicators commonly used to determine welfare. Damián et al. (2013) [175] aimed to determine which behaviors are triggered in lambs in response to separation from their mothers or changes in feeding and maternal separation during weaning. As a result, it was reported that separating lambs from their mothers is an important stressor and this plays a key role in the behavioral response to weaning in lambs. Separation from their mothers at weaning in grazing lambs was reported to increase the frequency of vocalization, walking, shadowing, as well as pacing behavior, and decrease the frequency of grazing. In addition, changes related to feeding and separation from adults are also an important component of the stress response to weaning. Casuriaga et al. (2022) [176] aimed to compare the behavioral and immunological responses of twin lambs housed with and without siblings after weaning. As a result, twin lambs housed with their siblings were more stressed at early weaning and showed more intense behavioral changes and poorer immunological status than twin lambs housed separately from their siblings.

Recent research suggests that age at weaning affects the development of ruminal bacteria in lambs at an early age [177]. Li et al. (2020) [178] reported that weaning significantly affected the morphological and functional development of the rumen and bacterial community composition. Wang et al. (2022) [169] reported that weaning affected rumen bacterial development and investigated rumen microbiota in early weaned lambs compared to conventionally weaned lambs. As a result, it was reported that early weaning caused a significant decrease in rumen microbiota richness and diversity in the short term and changes in rumen microbiota were associated with the persistence of weaning stress.

In conclusion, in traditional sheep production systems, lambs are suckled by their mothers and weaned at approximately 45–50 days of age. This situation involves a number of stress parameters for the offspring, both social, physiological, and nutritional, leading to a change in feeding regime. Weaning has a potential impact not only on the ewe–lamb relationship but also on the health of the lamb. In general, early weaned lambs involve more animal welfare concerns compared to conventionally weaned lambs, so it is important to develop strategies to minimize these responses.

## 8. Conclusions

Major changes have occurred in animal production in recent years. The rapid population growth in the world increases the demand for food- and production-based sectors, at the same time increasing the establishment of higher capacity production systems and the tendency towards integrated production activities. Nutritional conditions, housing conditions, and environmental conditions have changed significantly in this period and a dramatic increase in production has occurred. Issues such as sustainability of animal production under changing climatic conditions, evaluation of mutual interaction in different aspects, and ensuring adaptation in farm animals should be emphasized sensitively. All farm animals are exposed to some stress throughout their lives. In sheep, in addition to issues such as the type of stress, duration, degree of repetition, and how many different stress factors are present at the same time, many factors such as age, physiological period, breed, nutrition, yield status, health status, etc., have a role in the effects that will occur. Sheep try to cope with challenging conditions by using physiological and behavioral stress responses. When these responses are inhibited or not successful, typical behavioral and physiological symptoms of stress occur. In this situation, animal welfare is jeopardized and acute or chronic stress seriously affects animal health, productivity, and product quality of animal production. In this study, some important issues related to stress and its effects on sheep productivity (heat and cold stress, nutritional stress, transport stress, shearing stress, weaning stress) and relevant results obtained from some recently developed practices are discussed. Based on such research, information about the effects of management and husbandry requirements such as nutrition, housing, and environmental conditions on animal welfare is obtained and it provides ideas for making arrangements in these areas. As a result, stress research in sheep will contribute to the correct management of these processes, to produce solutions to different aspects of animal production with a holistic perspective, and to make improvements in these areas.

## Figures and Tables

**Figure 1 animals-13-02769-f001:**
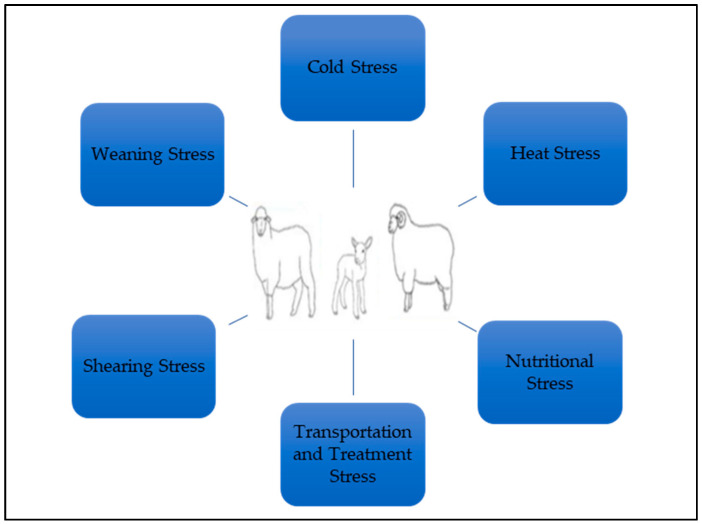
Some stress factors in sheep production.

**Figure 2 animals-13-02769-f002:**
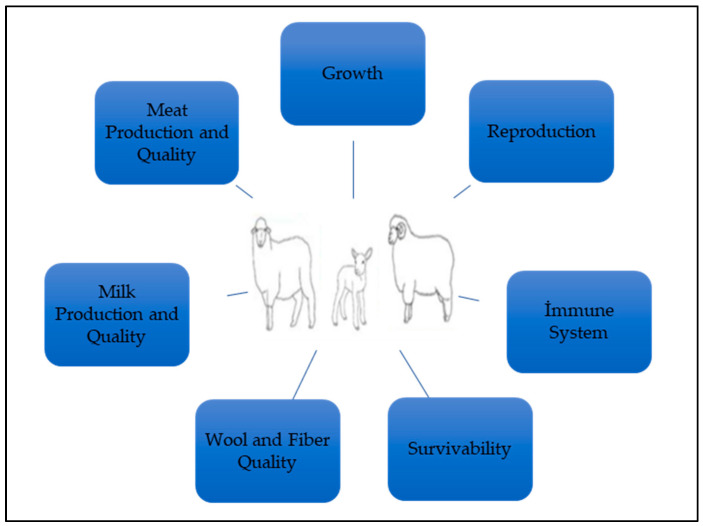
Some negative effects of stress factors on sheep productivity.

## Data Availability

Not applicable.

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
