# Peer review of "Stress Factors and Their Effects on Productivity in Sheep"

_animals, 2023, doi:10.3390/ani13172769_

Round 1

Reviewer 1 Report

Abstract: the length should be 200 words

Line 72: add a figure showing all factors that can cause stress

Add at paragraph 2 cold stress sub-paragraphs of Impact on Milk Production, Growth, Meat and Carcass Traits, Reproduction, Wool and Fibre Quality

Add at paragraph 3 sub-paragraphs of Impact on Wool and Fibre Quality

Add at paragraph 4 sub-paragraphs of Impact on Growth, Meat and Carcass Traits

Add a paragraph 8 where you analyze other animal breeding practices (numbering, grouping, tail docking, hoof care)

Add a paragraph 9 where you analyze inappropriate procedures, veterinary procedures (treatment, vaccination, blood tests, surgical intervention)

Add a paragraph 10 analyzing all blood parameters and molecular factors that change in cause of stress in sheep

References: rewrite following author’s guideline

Minor editing of English language required

Author Response

Dear Reviewer,

First of all, I would like to thank you for your suggestions and contributions to my article.

Necessary corrections and explanations have been made and presented in the supplementary files.

Thank you for your contribution to my article, best regards.

Abstract: the length should be 200 words

Necessary corrections have been made

Line 72: add a figure showing all factors that can cause stress

Necessary additions have been made.

Add at paragraph 2 cold stress sub-paragraphs of Impact on Milk Production, Growth, Meat and Carcass Traits, Reproduction, Wool and Fibre Quality

What is meant by paragraph 2 is not understood, it is not clear.

It is not clear what is meant by the paragraph 2 expression. However, if sub-headings were requested under the cold stress heading; There is not enough available work on this subject. For this reason, the existing studies have been presented in different paragraphs and a unity in the text has been created. Literature reports and results on the effects of cold stress in sheep are presented under a single title.

Add at paragraph 3 sub-paragraphs of Impact on Wool and Fibre Quality

What is meant by paragraph 3 is not understood, it is not clear.

However, if a subtitle (Impact on Wool and Fiber Quality) is requested under the heat stress heading; The relevant literature on this subject is included in a separate paragraph in the text. However, sufficient studies were not found to create a separate title. In addition, this subject has been evaluated as a separate topic while discussing nutritional stress.

Add at paragraph 4 sub-paragraphs of Impact on Growth, Meat and Carcass Traits

What is meant by paragraph 4 is not understood, it is not clear.

However, if it is requested to add a subtitle under the title of nutritional stress; In this title, the effects on growth are given in different parts. A separate subtitle was not deemed necessary.

Add a paragraph 8 where you analyze other animal breeding practices (numbering, grouping, tail docking, hoof care)

What is meant by paragraph 8 is not understood, it is not clear.

Add a paragraph 9 where you analyze inappropriate procedures, veterinary procedures (treatment, vaccination, blood tests, surgical intervention)

What is meant by paragraph 9 is not understood, it is not clear.

Add a paragraph 10 analyzing all blood parameters and molecular factors that change in cause of stress in sheep

What is meant by paragraph 10 is not understood, it is not clear.

References: rewrite following author’s guideline

References have been checked, corrections have been made.

Dear referee; It is not clear what paragraph 8-9-10 means in the corrections. However, if it is desired to open sub-headings for the factors mentioned, existing studies on these issues are included in the relevant paragraphs in the text and tried to be explained.

Of course, it is possible to increase the factors that cause stress in animals, these factors are also constantly updated with new studies. In this study, the main and most common stress factors in sheep breeding were tried to be mentioned. Otherwise, it is impossible to limit the scope of the study. In addition, there are certain limitations in the publication of articles (such as the number of pages). Therefore, in this study, many stress factors and their effects on productivity in sheep are included.

Reviewer 2 Report

This study attempts to explain the "Stress Factors and Effects on Productivity in Sheep." The study is within the scope of the journal. However, the information presented does not provide a full understatement of the consequences of certain stress factors on productive and reproductive traits. It is important to explain the physiological response; otherwise, the information presented is not novel.

I listed my other concerns below in the order I found them in the manuscript.

 Simple Summary

The simple summary is redundant and can be edited to improve clarity.

-       L8 – L9. Please define stress factors.

-       L8 – L9. Please edit the sentences as it is unclear.

Abstract

The abstract needs to be edited. The authors seem to mix the significance of "animal genetic improvement" and "breeding." The definition of breeding is "sexual reproduction that produces offspring."

-       L15 – L20. Do you mean animal genetic improvement? Sheep breeding per se does not improve those traits. Please edit.

-       L20 – L21. Environmental effects? It is not clear. Which constraints?

-       L21 – L23. Which stress factors? These factors have not been mentioned.

Introduction

The Introduction does not provide enough information to lead the reader to the "gap-knowledge" the authors want to address. It is essential to highlight the stress factors and how they impact the animal's well-being, production, and reproduction. Which is the mechanism by which these stress factors reduced productivity?

-       L33 – L34. Once again, the meaning of "sheep breeding" is wrongly used.

-       L36 – L40. I do not see the relevance of this sentence to the manuscript. It does not add anything.

-       L41 – L43. The sentence is very ambiguous. A better description of "external/internal/stressor" is needed.

-       L45 – L47. The reference is a review. Please provide the source for the sentence. This is a critical statement.

-       L54 – L58. What about over-nutrition? External factors such as noise (barking/screaming/music)? In addition, it is important to separate those factors that can have a short-term effect (shearing/hoovering) and those that can have a long-term effect with chronic consequences. Does blood collection cause stress?

Cold stress

The authors need to edit this section as the main objective is to present the physiological and productive consequences of cold stress. The authors ramble with the information presented, and the information does not flow correctly. A better structure and more literature support are needed.

-       L85 – L87. This line is out of context.

-       L87 – L89. These sentences need better literature support. It depends on so many different factors.

-       L91 – L92. This sentence needs better literature support.

-       L92 – L93. The authors are wrong. Better literature support is needed.

-       L127 – L138. Goats? I thought the manuscript was on sheep.

Heat Stress

A better understanding of the physiological response due to heat stress is needed.

-       L142 – L180. The structure does not flow smoothly.

-       L171 – L174. Sheep are one of the most efficient species in heat regulation.

-       L182 – L195. How heat stress reduced feed intake? Please explain the physiological response.

-       L196 – L201. How heat stress reduced milk production? Please explain the physiological response.

-       L237 – L251. How heat stress inhibited reproduction? Please explain the physiological response.

-       L254 – L264. This is a very interesting section. The authors need to address both under and over-nutrition. Moreover, in short-term nutrition, the animals can adapt their metabolism to their intake. A better explanation in this section is needed. Once again, the physiological response is needed.

-       L266 – L294. In this section, the authors address gestational nutrition. A better description of the maternal diet manipulation on fetal growth development is needed.

-       L296 – L308. Once again, a better description of the physiological response is needed.

-       L310 – L321. Once again, a better description of the physiological response is needed.

-       L320 – L321. How is this information relevant?

Shearing

There is so much literature on this topic. Please look up literature reported from Australia and New Zealand.

-       L399 – L453. If shearing causes an acute short-term stressor, how can it cause a reduction in production?

Weaning

This section needs to be edited. Indeed, weaning causes an acute short-term stressor; however, the evidence presented here is inconclusive. What is the physiological response of the progeny to the separation? Is there any epigenetic adaptation with detrimental consequences? Would it be possible to indicate the mortality rate due to weaning? Monetary loss as a consequence of weaning?

Some sections are difficult to understand in terms of flow.

Author Response

Dear Reviewer,

First of all, I would like to thank you for your suggestions and contributions to my article.

Necessary corrections and explanations have been made and presented in the supplementary files.

Thank you for your contribution to my article, best regards

Simple Summary

The simple summary is redundant and can be edited to improve clarity.

-       L8 – L9. Please define stress factors.

-       L8 – L9. Please edit the sentences as it is unclear.

-     A simple summary has been corrected in line with the suggestions.

Abstract

The abstract needs to be edited. The authors seem to mix the significance of "animal genetic improvement" and "breeding." The definition of breeding is "sexual reproduction that produces offspring."

-       L15 – L20. Do you mean animal genetic improvement? Sheep breeding per se does not improve those traits. Please edit.

-       L20 – L21. Environmental effects? It is not clear. Which constraints?

-       L21 – L23. Which stress factors? These factors have not been mentioned.

 -  Necessary corrections were made in the text. The word "breeding" that caused the misunderstanding has been corrected with the appropriate wording.

Introduction

The Introduction does not provide enough information to lead the reader to the "gap-knowledge" the authors want to address. It is essential to highlight the stress factors and how they impact the animal's well-being, production, and reproduction. Which is the mechanism by which these stress factors reduced productivity?

 -       L33 – L34. Once again, the meaning of "sheep breeding" is wrongly used.

-    Necessary corrections have been made in the text.

-       L36 – L40. I do not see the relevance of this sentence to the manuscript. It does not add anything.

-     The text has been edited, but not completely removed.

-       L41 – L43. The sentence is very ambiguous. A better description of "external/internal/stressor" is needed.

-      The sentence has been rewritten in the text.

-      L45 – L47. The reference is a review. Please provide the source for the sentence. This is a critical statement.

-      Literature has been added.

-       L54 – L58. What about over-nutrition? External factors such as noise (barking/screaming/music)? In addition, it is important to separate those factors that can have a short-term effect (shearing/hoovering) and those that can have a long-term effect with chronic consequences. Does blood collection cause stress?

 -    Of course, the factors mentioned have many effects on animals. However, the factors mentioned in the text came to the fore and became the subject of more studies. Therefore, it was not possible to address all of the factors affecting stress in the article.

Cold stress

The authors need to edit this section as the main objective is to present the physiological and productive consequences of cold stress. The authors ramble with the information presented, and the information does not flow correctly. A better structure and more literature support are needed.

-       L85 – L87. This line is out of context.

-     This paragraph is included because of the indirect effects of the change in the pastures in winter and the feed intake.

-       L87 – L89. These sentences need better literature support. It depends on so many different factors.

-     Editing has been done in the text, it is compatible with the literature.

-       L91 – L92. This sentence needs better literature support.

-    The information presented in the text is in harmony with the literature reports.

-       L92 – L93. The authors are wrong. Better literature support is needed.

-    The information presented in the text is in harmony with the literature reports.

-       L127 – L138. Goats? I thought the manuscript was on sheep.

-     Corrections have been made in the text.

Heat Stress

A better understanding of the physiological response due to heat stress is needed.

-       L142 – L180. The structure does not flow smoothly.

-     In the parts mentioned in the text, the direct and indirect effects of heat stress have been tried to be explained. For this reason, many factors have been tried to be mentioned in the text in a way that does not disturb the flow.

-       L171 – L174. Sheep are one of the most efficient species in heat regulation.

-    Dear referee, I agree with your opinion, it is stated in the text that sheep are advantageous compared to other species. However, heat stress causes many negative effects on sheep. This is tried to be stated here.

-       L182 – L195. How heat stress reduced feed intake? Please explain the physiological response.

-    Necessary additions have been made to the text.

-       L196 – L201. How heat stress reduced milk production? Please explain the physiological response.

   Necessary explanation is available in the text.

-       L237 – L251. How heat stress inhibited reproduction? Please explain the physiological response.

   In the relevant paragraph presented in the text, its effects on reproduction in female and male animals are explained, and literature reports are also presented.

-       L254 – L264. This is a very interesting section. The authors need to address both under and over-nutrition. Moreover, in short-term nutrition, the animals can adapt their metabolism to their intake. A better explanation in this section is needed. Once again, the physiological response is needed.

-    Studies on both undernutrition and overnutrition under nutritional stress are included in the relevant sections.

-       L266 – L294. In this section, the authors address gestational nutrition. A better description of the maternal diet manipulation on fetal growth development is needed.

-    In the text, studies and literature reports on this subject are presented.

-       L296 – L308. Once again, a better description of the physiological response is needed.

-     There are related studies in the text, but some additions have also been made.

-       L310 – L321. Once again, a better description of the physiological response is needed.

-     Necessary literature reports are available in the text.

-       L320 – L321. How is this information relevant?

-    It was written because it is a study on feeding in sheep and has a positive effect on reproduction. It can be removed if necessary.

Dear referee, it has been requested to explain the physiological effects of stress factors in this area in general. There are physiological effects of stress factors present in detail in the relevant literature reports in the text. Some additions have been made. However, it is not possible to take all of these highly complex and detailed effects separately and to explain them in this study. The explanation of each factor is a separate study. Therefore, no further details were included in the text.

Shearing

There is so much literature on this topic. Please look up literature reported from Australia and New Zealand.

-       L399 – L453. If shearing causes an acute short-term stressor, how can it cause a reduction in production?

-    Dear referee, you are right, there are many studies in this field. Many studies are included in the text and the results are shared.

Weaning

This section needs to be edited. Indeed, weaning causes an acute short-term stressor; however, the evidence presented here is inconclusive. What is the physiological response of the progeny to the separation? Is there any epigenetic adaptation with detrimental consequences? Would it be possible to indicate the mortality rate due to weaning? Monetary loss as a consequence of weaning?

Studies have shown that weaning causes stress in animals. Literature reports are reported in the text. Many different studies are included and the results are shared.

Round 2

Reviewer 1 Report

add three further paragraphs to the text, which will be 8-9-10

8 where you analyze other animal breeding practices (numbering, grouping, tail docking, hoof care)

9 where you analyze inappropriate procedures, veterinary procedures (treatment, vaccination, blood tests, surgical intervention)

 10 analyzing all blood parameters and molecular factors that change in cause of stress in sheep

Minor editing of English language required

Author Response

Dear Referee,

First of all, I would like to thank you for your suggestions and contributions to my article.

Necessary correction is presented with explanation.

Thank you for your contribution to my article, best regards.

add three further paragraphs to the text, which will be 8-9-10

8 where you analyze other animal breeding practices (numbering, grouping, tail docking, hoof care)

9 where you analyze inappropriate procedures, veterinary procedures (treatment, vaccination, blood tests, surgical intervention)

10 analyzing all blood parameters and molecular factors that change in cause of stress in sheep

Dear referee, 3 more paragraphs were requested to be added to the text. (8- numbering, grouping, tail docking, hoof care, 9- treatment, vaccination, blood tests, surgical intervention, 10- analyzing all blood parameters and molecular factors that change in cause of stress in sheep)

Dear referee, there is not enough literature in sheep to be able to create separate paragraphs for the 3 paragraph titles that are requested to be added to the article.

There is not enough available work on this subject. For this reason, the existing studies have been presented in different paragraphs and a unity in the article has been created.

The limited existing studies on these subjects have been tried to be included under other existing titles in the article.

Of course, it is possible to increase the factors that cause stress in animals, these factors are also constantly updated with new studies. In this study, the main and most common stress factors in sheep breeding were tried to be mentioned. Otherwise, it is impossible to limit the scope of the study. Therefore, in this study, many stress factors and their effects on productivity in sheep are included.

Reviewer 2 Report

The manuscript has improved.

The only observation is that the conclusion is too long. It can be shortened.

Author Response

Dear Referee,

First of all, I would like to thank you for your suggestions and contributions to my article.

Necessary correction is presented with explanation.

Thank you for your contribution to my article, best regards.

The only observation is that the conclusion is too long. It can be shortened.

Thank you for your suggestion, the conclusion has been tried to be presented in the most concise form possible.
